## REVIEW ARTICLE

# Melanopsin-mediated optical entrainment regulates circadian rhythms in vertebrates

Deng Pan[1], Zixu Wang[1], Yaoxing Chen[1] & Jing Cao [1✉]

Melanopsin (OPN4) is a light-sensitive protein that plays a vital role in the regulation of circadian rhythms and other nonvisual functions. Current research on OPN4 has focused on mammals; more evidence is needed from non-mammalian vertebrates to fully assess the significance of the non-visual photosensitization of OPN4 for circadian rhythm regulation. There are species differences in the regulatory mechanisms of OPN4 for vertebrate circadian rhythms, which may be due to the differences in the cutting variants, tissue localization, and photosensitive activation pathway of OPN4. We here summarize the distribution of OPN4 in mammals, birds, and teleost fish, and the classical excitation mode for the non-visual photosensitive function of OPN4 in mammals is discussed. In addition, the role of OPN4-expressing cells in regulating circadian rhythm in different vertebrates is highlighted, and the potential rhythmic regulatory effects of various neuropeptides or neurotransmitters expressed in mammalian OPN4-expressing ganglion cells are summarized among them.

In the vertebrate retina, the sensitivity of dim-light vision is supported by rod photoreceptors, whereas cone photoreceptors mediate color discrimination and high visual acuity at higher light intensities[1,2]. Compared with visual forming visual pathways, the regulation of non-image forming visual pathways is performed by intrinsically photosensitive retinal ganglion cells (ipRGCs), such as circadian entrainment[3], pupillary light reflex[4–6], and time-restricted feeding[7]. Although ipRGCs are less represented in mammalian retina (mice: ~1–5%, human: ~0.4–1.5%)[8–12], melanopsin (OPN4), as an opsin, gives it powerful non-image forming function[13,14].

OPN4 is a G protein-coupled receptor initially identified in the dermal melanocytes of *Xenopus laevis*. It includes an extracellular amino-terminal and seven transmembrane domains with high homology to invertebrate opsins[15]. The OPN4 gene has been detected in most vertebrates and analyzed in two lineages, xenopus (OPN4x) and mammalian (OPN4m) orthologs[16]. It was also found that there are two OPN4m splice variants in mice and humans, the short (OPN4-S) and long (OPN4-L) isoforms, which differ mainly in the number of phosphorylatable serines and threonines in the C-terminus, which may lead to differences in the inactivation dynamics of OPN4 in different species[16,17]. Regarding photosensitivity, the $\lambda_{max}$ (peak sensitivity) of OPN4 was 480 nm as measured directly in light response to ipRGCs, confirmed by mouse models lacking rods and cones[13,18,19]. Notably, the peak sensitivity of OPN4 shows some minor differences in many studies depending on differences in the detection methods, technology, or species[20,21].

Here, we collate the distribution of OPN4 in mammals, birds, and teleost fish based on published evidence. We then highlight the mechanisms by which the non-visual photosensitization of OPN4 mediates in vertebrate circadian rhythm regulation. Taking the photosensitive activation of OPN4 as a starting point, our review focuses on the mechanism of OPN4-mediated photoentrainment action in circadian rhythm regulation in vertebrates. Admittedly, the other OPN4 activations of G-protein coexist and have been summarized in

[1] Laboratory of Anatomy of Domestic Animals, National Key Laboratory of Veterinary Public Health and Safety, College of Veterinary Medicine, China Agricultural University, Haidian, 100193 Beijing, China. ✉email: caojing315@126.com

recent relevant reviews[22–24]. In this article, the G$_{q/11}$ pathway in the OPN4-mediated phototransduction was mainly described due to its widespread presence in vertebrate ipRGCs[25–28]. In addition, OPN4-mediated light entrainment impacts melatonin secretion in the vertebrate retina and pineal gland. This leads to more diverse rhythmic regulatory pathways in non-mammalian vertebrates than mammals.

## Distribution of OPN4 in vertebrates

**Mammalian**. Mammalian OPN4 is derived from a single OPN4 gene with two splice variants in ipRGCs, which has been localized and accurately classified by much evidence. OPN4 is also distributed in mammalian peripheral tissues (Table 1), but its functions remain to be further investigated. Therefore, OPN4 in mammalian ipRGCs will be discussed first.

The OPN4-expressing ipRGCs were previously thought to be a homogeneous cell population with sparsely branched dendritic trees on the outermost layer of the inner plexiform layer in mammals[8]. Subsequently, the expression of OPN4 in M1-M6 ipRGCs in the mouse retina has been identified[12,29]. This OPN4 in ipRGCs with various morphological and physiological characteristics can provide complete light-dark discrimination and partial vision in rodless/coneless (rd/rd cl) mice[30,31]. Among these subtypes of ipRGCs, M1 expressed the highest content of OPN4, and it mainly exerts OPN4-induced photoentrainment[32–34]. Correspondingly, the suprachiasmatic nucleus (SCN) is innervated primarily by M1-subtype ipRGCs (~80%), and OPN4 in M1-subtype ipRGCs significantly regulates rhythmic regulation in mammals[35,36].

**Birds**. Mammals lost OPN4x during evolution and chromosomal re-arrangements[37,38], which accompanied mammal adaptation to the nocturnal niche[39,40]. In a bird's retina, two lineages for OPN4 are expressed[41]. OPN4m is stably expressed in the retinal ganglion cells (RGCs) during the development of birds, whereas OPN4x was limited to the forming RGCs at embryonic 8 (E8), but mainly expressed in PROX1-positive horizontal cells (HCs) at E15[42]. These OPN4-expressing horizontal cell precursors continue to express OPN4x after migrating and developing into horizontal cells[43].

In contrast to mammals, the distribution of OPN4 in birds is no longer concentrated in the retina (Fig. 1)[44]. Bird pinealocytes are directly photosensitive[45], and the reconstitution of the recombinant proteins with 11-*cis*-retinal demonstrated that it expresses two lineages of melanopsins[46]. The transcriptional levels of OPN4 in the pineal gland showed a more robust diurnal feature than that in the retina and were significantly increased at night[41]. Although avian pinealocytes possess both OPN4m and OPN4x (also called OPN4-1 and OPN4-2 in chickens), their distribution is not cell-specific. It may activate different types of G proteins to perform light-sensing functions[47]. In addition, multiple nuclei composing deep brain photoreceptors in birds also express OPN4 (Fig. 1), including the lateral septal organ,

premammillaris nucleus, paraventricular nucleus (PVN), and paraventricular organ[48]. OPN4-positive dopaminergic neurons in these nuclei can respond to daytime length[49].

**Reptiles**. The studies on OPN4 in reptiles has mainly focused on lizards, sea snakes, and turtles, but there still needs to be more evidence to locate the expression sites of OPN4 and its isoforms accurately. To date, OPN4m was not detected in sea snakes, while OPN4x was mainly expressed in RGCs and cone cells[50]. Although OPN4x-positive staining was also observed in the inner nuclear layer[50], the cell type could not be determined. In freshwater turtles, OPN4m is highly expressed in the retina, but it is not yet certain whether OPN4m is localized in RGCs[51]. In extraretinal photoreceptors, OPN4x is also expressed in the lateral eye and brain of ruin lizards but has not been detected in the pineal gland[52].

**Amphibians**. When it was discovered, melanopsin was found in the retina, melanophores, and deep brain photoreceptors of *Xenopus laevis*[15]. Both OPN4m and OPN4x have been localized in RGCs, horizontal cells, and pineal complex[53,54]. OPN4-expressing RGCs have been shown to participate in the melanocyte pigmentation process by producing alpha-melanocyte stimulating hormone in the pituitary gland[55]. OPN4 in the pineal complex may participate in the change of skin color through the neuroendocrine pathway[54]. These photosensitive neuroendocrine circuits enable Xenopus to maintain rapid physiological pigmentation change.

**Teleost fish**. Five splice variants were detected in zebrafish (OPN4.1, OPN4a, OPN4b, OPN4xa, and OPN4xb), which are derived from two melanopsin lineages (OPN4m and OPN4x) to confer overall photosensitivity to the teleost retina and to adapt to the dynamic light environments in the aquatic habitats[56]. Similar to birds, both lineages of OPN4 are expressed in RGCs, and partial OPN4 splice variants are distributed in horizontal cells[57–59], which independently mediates the role of HCs in photosensitive signaling[60]. In extraretinal tissue, OPN4m was detected in the dorsal thalamus, ventral hypothalamus, and nucleus lateralis tuberis pars lateralis; OPN4x was evident in the SCN and habenular nucleus[59]. Evidence for functional partitioning suggests that OPN4m mediates the light-seeking behavior in larvae distributed in the preoptic area[61], whereas OPN4x regulates circadian rhythms in the SCN[62]. The zebrafish pineal gland is a photosensitive structure with various opsins, a subpopulation of pinealocytes capable of sensing shorter wavelength light, characterized by the expression of OPN4x[63]. In addition, two splice variants of OPN4, OPN4.1 and OPN4xb, were detected in the pineal gland, which is responsible for inhibiting melatonin synthesis during the day and maintaining voluntary movements in a state of absolute arousal[64].

Overall, current evidence has shown that OPN4 is mainly distributed in the retina of mammals, while it is also widely

**Table 1 Photosensitivity of vertebrate OPN4 in peripheral organs.**

| Taxa | Species | Location | Related effects | References |
|---|---|---|---|---|
| Amphibia | Xenopus laevis | Melanocyte | Skin pigmentation | Provencio et al.[15] |
| Reptile | Hydrophiinae | Skin | Tail phototaxis | Crowe-Riddell et al.[27] |
| Mammal | Mouse | Aortas, pulmonary arteries, airway smooth muscle | Light-dependent relaxation | Sikka et al.[172]; Barreto et al.[173]; Yim et al.[174] |
| Mammal | Mouse | Melanocytes | Pigmentation | de Assis et al.[175] |
| Mammal | Human | Mesenchymal stem cells | Angiogenesis | Yang et al.[176] |
| Mammal | Human | Subcutaneous white adipose tissue | Lipolysis of lipid droplets | Ondrusova et al.[177] |

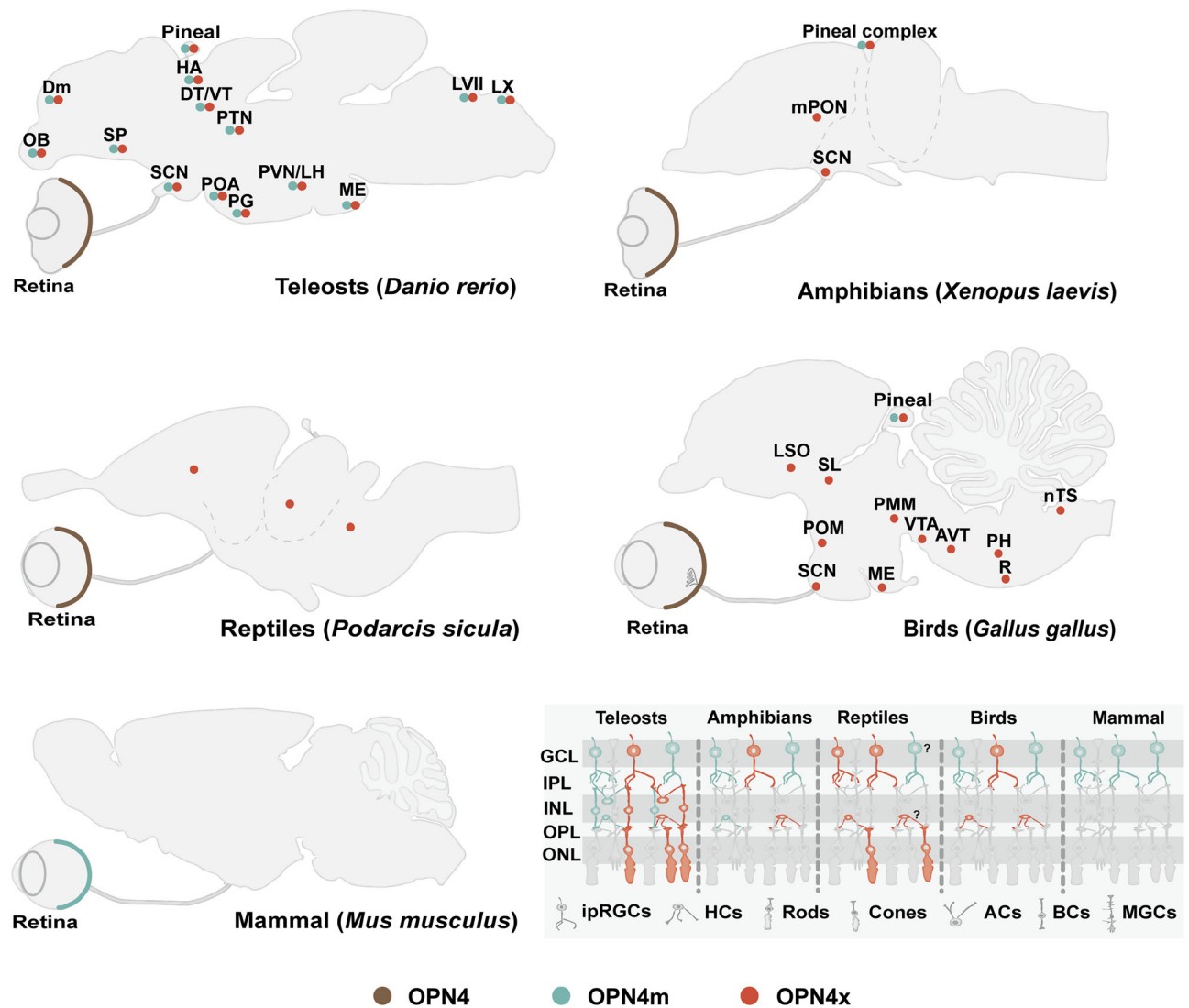

**Fig. 1 Distribution of OPN4 in teleost fish, amphibian, reptiles, birds, and mammals on generalized sagittal sections.** In teleost fish and birds, two orthologs of OPN4 are distributed in the retina, brain, and pineal gland[44,61,64,166–168]. For teleosts, amphibians, and reptiles, splice variants of OPN4 are classified as OPN4m and OPN4x. In the reptile brain, OPN4x expression has been detected in the telencephalon, mesencephalon, and rhombencephalon, but the specific nuclei are still unclear. It is important to note that the available evidence does not determine the cell type of OPN4x in the inner nuclear layer or whether OPN4m is present in RGCs in reptiles. Mammalian OPN4 is mainly expressed in the retina, which integrates more complex photosensitive functions and widely projects to different brain regions through different subtypes of ipRGCs to regulate various physiological functions[14]. Brackets indicate representative species. AC amacrine cell, AVT area ventralis of tsai, BC bipolar cell, Dm the medial zone of the dorsal telencephalic region, DT dorsal thalamus, HA habenula, HC horizontal cell, ipRGC intrinsically photosensitive retinal ganglion cell, LH lateral hypothalamic nucleus, LSO lateral septal organ, LVII facial lobe, LX vagal lobe, ME median eminence, MGC Muller glial cell, mPON magnocellular preoptic nucleus, nTS nucleus tractus solitarius, PG preglomerular area, PH plexus of horsley, PMM nucleus premammillaris, POA preoptic area, POM medial preoptic nucleus, PTN posterior tuberal nucleus, PVN periventricular nucleus, R raphe nucleus, SCN suprachiasmatic nucleus, SL nucleus septalis lateralis, SP subpallium, VT ventral thalamus, VTA ventral tegmental area.

expressed in the brains of teleost fish, amphibians, reptiles, and birds (Fig. 1).

### Light activation of OPN4 in the retina

OPN4 is a G protein-coupled receptor with 11-*cis* retinal as a covalently bound protonated Schiff base (PSB11)[65]. Under the induction of light, the conformation of 11-*cis* retinal changed with the transformation of PBS11 to its all-*trans* isomer, which changed the state of PSB11 to a 7-*cis* state[66–68]. In this series of changes, the 11-*cis* and 7-*cis* retinal indicate OPN4's silent state, while the all-*trans* structure indicates light signaling conversion[68]. This tristability confers on OPN4 a sustained response to light and a broader spectrum of its own[69]. Following this reaction, the

$G_{q/11}$ class of G-proteins will become active and further trigger the activation of phospholipase C-beta 4 (PLCβ4). This leads to the hydrolysis of phosphatidylinositol 4,5-bisphosphate to form inositol triphosphate and diacylglycerol through the transient receptor potential cation channel subfamily C member 6/7 (TrpC6/7) nonselective cation channels in the cell membrane and finally increases the intracellular $Ca^{2+}$ concentration (Fig. 2)[14,21,23]. Using calcium ion probes, Sekaran et al. consistently found that OPN4 can specifically respond to a wavelength of 470 nm with a significant increase in $Ca^{2+}$ concentration in a mouse model lacking cone and rod photoreceptors[70]. In addition, a recent study demonstrated that internally released $Ca^{2+}$ marks the opening of the OPN4-

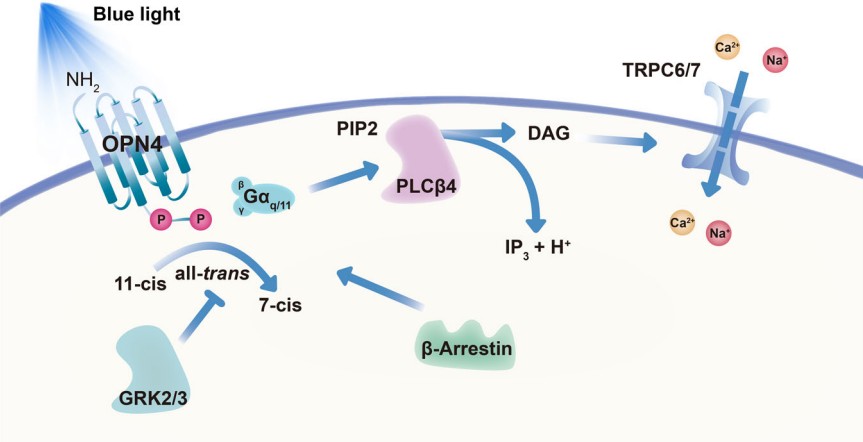

**Fig. 2 Activation and termination of OPN4 in M1-subtype ipRGCs of mammals.** The OPN4-mediated light-sensitive pathways are predominantly triggered by the downstream G$_{q/11}$, PLCβ4, and TRPC6/7 cation channels in mammals. Retinaldehyde is covalently bonded to the transmembrane structure in OPN4, and light (especially near 480 nm) can change its conformation from an 11-cis to an all-trans state to a 7-cis state (silent state). It will trigger downstream G$_{q/11}$ coupling, causing PLCβ4 to break down PIP2 into DAG and IP3, where DAG activates the opening of the TRPC6/7 cation channels. The activated C-terminus of OPN4 is phosphorylated in response to GRK2/3, resulting in inactivation. This process may also involve β-Arrestin 2. In addition, β-Arrestin 1 leads to the isomer regeneration of OPN4, which serves subsequent light activation. DAG diacylglycerol, G$_{q/11}$ G protein subunit alpha q/11, GRK2/3 G protein-coupled receptor kinase 2/3, PLCβ4 phospholipase C-beta 4, PIP2 phosphatidylinositol bisphosphate, IP3 inositol triphosphate, TRPC6/7 transient receptor potential cation channel subfamily C member 6/7.

mediated light-sensitive pathway[71], which is the opposite of the hyperpolarization of rods and cones[14]. In the opsin photosensitive response termination, OPN4 is subject to C-terminal phosphorylation. Its phosphorylation process preferentially interacts with G protein-coupled receptor, kinase 2/3 (GRK2/3), preventing OPN4-expressing ipRGCs from generating sustained action potentials after light stimulation[72–74]. Meanwhile, arrestin is also involved in the inhibition and reactivation of the light response of OPN4. When the C-terminus of OPN4 is phosphorylated, it can bind to arrestin. β-arrestin 2 primarily regulates the deactivation of OPN4, whereas β-arrestin 1 initiates regeneration of OPN4[75,76]. The above responses allow ipRGCs to sustain responses under prolonged illumination (Fig. 2).

Due to the complexity of the G protein family and the variation of OPN4 subtypes in different species, the optical signal transduction of the OPN4 pathway is mainly dependent on G$_{q/11}$ in M1-subtype ipRGCs in mammalian and partial OPN4-expressing cells in non-mammalian vertebrates[25,28,47,77,78]. Recent studies have shown that adenylyl cyclase 2 and cAMP mediate the phototransduction of OPN4 in M4-subtype ipRGCs[24]. Considering the ability of retinal adenosine to influence photosensitive electrophysiological activity in the retina[79], the effect of cAMP on OPN4 phototransduction cannot be ignored. Admittedly, OPN4-mediated phototransduction mechanisms have also been implicated in species that involve G$_{i/o}$ (human, mouse, and amphioxus), Gs (chicken), or Gt (chicken) activity[26,47,80].

**Photosensitive regulation of circadian rhythms by OPN4**

OPN4-induced non-visual photosensitive signals can target numerous nuclei, including the SCN, intergeniculate leaflet, ventral lateral geniculate nucleus, and olivary pretectal nucleus (OPN)[81]. The SCN is the center for orchestrating mammalian circadian rhythms, while the OPN is essential for regulating the pupillary light reflex[82,83]. The synaptic structures at the ends of these tracts specialize in different nucleus regions, leading to differences in threshold sensitivity, speed, and accuracy of visual responses in these nuclei[84]. As we mentioned above, M1-subtype ipRGCs form the core of non-visual photosensitivity. According to the molecularly defined Brn3b transcription factor expression,

the M1 subtype consists of two distinct subpopulations. The majority of projections from M1 ipRGCs to the thalamus and midbrain are Brn3b-positive M1-subtype ipRGCs[85], which regulate OPN4-dependent pupillary light reflexes and light-induced acute body temperature changes[83,86,87]. The SCN is innervated by Brn3b-negative M1-type ipRGCs[83], and it is here that the SCN orchestrates multiple oscillators with a duration of almost 24 h[88,89]. Therefore, the non-visual function of OPN4 contributes to controlling central rhythms in mammals via Brn3b-negative M1-subtype ipRGCs projections in the SCN region.

Unlike mammals, chick retinal ganglion cells were classified into six subgroups according to their somal and dendritic characteristics (subgroups Ic, Is, IIc, IIs, IIIs, and IVc)[90,91]. The subgroups IIs and IIIs had a more significant proportion of thalamic projection[92]. Identifying the function of RGCs from chickens is still challenging, despite similarities in RGC projection pathways to the brain between birds and mammals. Brn3b molecular markers commonly used in mammals may not be suitable for birds. All types of Brn3 factors (Brn3a, Brn3b, and Brn3c) can promote the differentiation of chick RGCs and are not mainly regulated by Brn3b as in mammals[77]. Furthermore, species differences make it challenging to directly administer current antibodies and viral vectors to birds' retinas or central nervous systems. Nevertheless, studying retinal ganglion cell subtypes in chicks may be more effectively accomplished using in vivo transfection or electroporation transfection[93–95]. For non-mammalian vertebrates, the pineal gland of birds and teleost fish has rhythmic pacing functions and is involved in constituting the multi-oscillatory circadian timing system[96,97]. The photosensitization of photoreceptors in the retina by OPN4 may have limited effects on circadian rhythms in these species. Therefore, when discussing OPN4-mediated non-visual photosensitive functions, the pineal gland of non-mammalian vertebrates will also be emphasized.

It should be noted that cones and rods can affect not just the local biological clock of the retina[98,99], but also the master clock of the SCN[100]. During development, ipRGCs form functional connections with the cone/rod system in the inner reticular layer, allowing them to serve as relays to transmit collected rod and cone information to the brain while retaining their intrinsic

photosensitivity[30,101]. Photoentrainment induced by rods can influence the master clock via cone circuits, which may complement the function of photoentrainment in ipRGCs in dim light[102]. Accordingly, the light power required to activate OPN4 (>1 μW) under in vitro conditions is higher than conventional retinoids (~0.2 μW)[103]. At the same time, ultraviolet ($\lambda_{max}$ 365 nm) and green ($\lambda_{max}$ 505 nm) sensitive cone cells are also able to indirectly influence the electrophysiological activity of the neurons in the SCN via ipRGCs, contributing to photoentrainment[100,104]. Additionally, harmonizing the photosensitive signals from the cones, rods, and ipRGCs also plays a crucial role in ensuring the pupillary light reflex functions properly[105]. Thus, the influence of cones and rods on circadian rhythm regulation should not be undervalued.

**Contribution of OPN4 to mammalian circadian rhythms**. Exposure to monochromatic blue light (460 nm) can suppress human melatonin levels and interfere with resetting circadian rhythm[106,107]. As part of this regulation, the photosensitive signal of OPN4 is first transmitted to the SCN through the retinohypothalamic tract (RHT), followed by the paraventricular nucleus and the intermediolateral nucleus via the polysynaptic circuit distributed in the SCN region, and finally to the release of melatonin innervated by the sympathetic nerve in the superior cervical ganglion (SCG)[108,109]. In addition to this approach, OPN4-positive ipRGCs can rely on self-synthesized neurotransmitters and neuropeptides to more directly and rapidly affect the SCN master clock.

Retinal glutamatergic signals are responsible for transmitting external light information to the SCN, and binocular enucleation induced a significant decrease in vesicular glutamate transporter 2 (Vglut2) immunoreactivity in the ventrolateral part of the SCN[110]. The experiments in *OPN4^Cre/+^::Vglut2^flox/flox^* transgenic mice proved that the glutamate transmission from ipRGCs is necessary for light to entrain circadian rhythms in dim light[111]. Regarding synaptic connections, glutamatergic ipRGCs have neural projections with many photosensitive neurons in the SCN. Many Vglut2-immunoreactive axons were observed to be in synaptic contact with vasoactive intestinal peptide (VIP)- and gamma-aminobutyric acid (GABA)-positive neurons[112]. ipRGCs have direct synaptic connections with arginine vasopressin (AVP) neurons in the dorsal SCN[113]. Glutamatergic signaling primarily controls the expression of clock genes concerning the regulation of the SCN master clock. The glutamatergic activation of the N-methyl-D-aspartic acid (NMDA) receptor leads to an influx of extracellular $Ca^{2+}$, followed by $Ca^{2+}$/calmodulin-dependent kinase II and nitric oxide synthase activation[114,115]. Then, the increased nitric oxide levels activate ryanodine receptors (RyRs) in the intracellular endoplasmic reticulum[116]. Finally, intracellular $Ca^{2+}$ is released by activated RyR, leads to phosphorylation of cAMP response element-binding (CREB) protein, and regulates transcription of period and cryptochrome by CLOCK and BMAL[117]. During the maintenance of the circadian rhythm, the transcription factor CREB can integrate photosensitive information and mediate the reset of the circadian rhythm[118]. It is undeniable that the strength of this OPN4-mediated glutamatergic signaling is different in species with diurnal activity patterns, which is also reflected in their nonidentical phase-response curves (PRC). The projection of ipRGCs-SCN in the Nile rat (*Arvicanthis niloticus*) is comparable to that of the Syrian hamsters[119]. However, there are differences in sensitivity to phase movement between the two species on the NMDA-induced PRC[120,121], which is also reflected in the strong resistance of *Arvicanthis niloticus* to NMDA[122].

ipRGCs also express a peptide neurotransmitter called pituitary adenylate cyclase-activating peptide (PACAP) and colocalize with glutamate at the terminals of RHT in the SCN[123,124]. Previous studies have shown that adding PACAP to SCN slices in wild mice at circadian time (CT) 6 can advance the peak of the SCN activity rhythm in this and subsequent circadian rhythms[125]. However, the phase and amplitude of the neuronal firing rhythm do not change in *Adcyap1* (adenylate cyclase activating polypeptide 1, encoding PACAP) knockout mice at CT6 and CT7 in the SCN[126]. In addition, light stimulation in the early night (CT15) delayed the phase, while light stimulation in the late night (CT21) advanced the phase[127]. Consistently, the influence of PACAP on the circadian rhythm depends on glutamate in the late night (phase advance), and the independent regulation of circadian rhythms by glutamate occurs in the early night (phase delay)[126,128,129]. The time-dependence phase shift at night may be due to PACAP and glutamate acting on different SCN neuronal subpopulations. Compared to glutamate, the positive signals for PACAP were mainly distributed in the dorsomedial SCN and a small amount in the central/ventral SCN[126,130]. Using c-Fos to mark neuronal activity, neurons with significant light responses during the subjective daytime were distributed in the dorsal SCN, and light did not affect the rhythm phase of mice[131]. This phenomenon is consistent with the evidence that circadian rhythms are not altered in *Adcyap1* knockout mice. Regarding regulatory mechanisms, PACAP has a regulatory effect on glutamatergic calcium signaling and has a different time window from glutamate in regulating CREB phosphorylation[132,133]. PACAP can regulate circadian rhythm by differentially regulating mitogen- and stress-activated protein kinase 1 phosphorylation downstream of p42/44 mitogen-activated protein kinase between day and night[134]. Therefore, in terms of the autonomous rhythm of SCN neurons, PACAP may be a supplementary factor to OPN4-mediated SCN mastering circadian clock rhythms in response to the risk of potential circadian imbalance underlying Vglut2 deficiency when glutamate stimulation alone is insufficient.

Most OPN4-containing cells also expressed vasopressin (VP), which has glutamatergic nerve fibers projecting to the non-visual nuclei of the brain, and the application of VP receptor antagonist decreases the response of SCN neurons to photic entrainment of the RHT[135]. Additionally, vasopressinergic axons can affect the activity of ventral SCN cells in a VP-dependent manner[136]. Applying the antagonists of vasopressin V1a and V1b receptors to the SCN can promote (near instantaneous) re-entrainment to the new light/dark cycle[137]. Current evidence suggests that VP+ ipRGCs have synaptic co-localization with gastrin releasing peptide (GRP)- and VIP-positive neurons, but VP+ ipRGCs are not directly connected in AVP neurons[135]. According to single-nucleus RNA sequencing assays, all AVP clusters expressed glutamate receptor subunits with minimal expression of GABA receptors. However, some of their AVP nonlight-responsive clusters could express VIP receptor type 2[138]. Therefore, we speculate that photosensitive AVP, GRP, and VIP neurons may be downstream neurons of ipRGCs when OPN4-expressing ipRGCs secrete both glutamate and VP.

The effect of GABA on the master clock can be excitatory or inhibitory in different contexts, but there is no doubt about its importance[139,140]. The light information transmitted by ipRGCs can induce oscillations in the GABAergic system in the SCN. This may be because GABA_B receptors are highly localized ventral to the SCN and are closely related to the signal afferents and the terminal synaptic remodeling of RGCs[141]. Studies in hamsters have shown that the antagonism of either GABA_A (ionotropic) or GABA_B receptors (metabotropic) in the SCN significantly

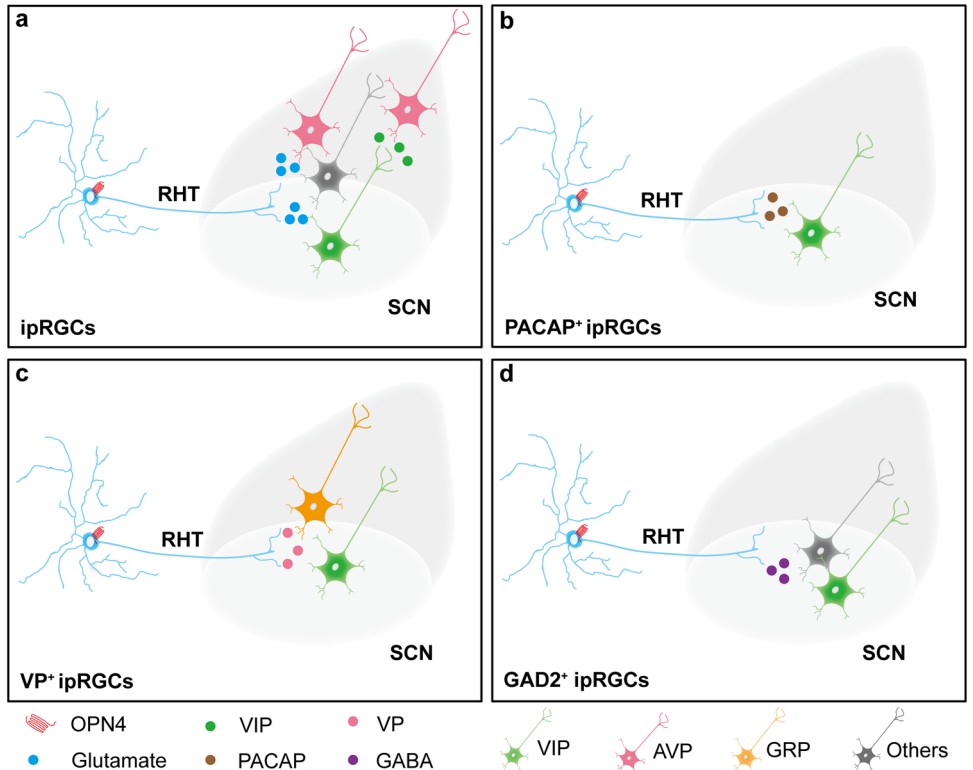

**Fig. 3 The light entrainment of OPN4 on circadian rhythms may involve multiple neural projection pathways, including neurotransmitters or neuropeptides.** ipRGCs are a class of retinal ganglion cells that express OPN4 (red) and can transmit OPN4-mediated photosensitive signals to the SCN via RHT projections. **a** Vglut2, but not Vglut1, packages glutamate (solid blue circles) into synaptic vesicles in these axons. These ipRGCs axons mainly make synaptic contacts with VIP neurons (green), AVP (pink), and other light-responsive neurons (gray) in the SCN[169,170]. The VIP neurons form part of the SCN core region and may communicate with AVP neurons via VIP receptor type 2. **b** Some ipRGC axons can also release PACAP (solid brown circles) to regulate VIP neurons via VPAC$_2$ and PAC$_1$ receptors[171]. In addition, some OPN4-expressing ipRGCs also expressed VP (solid pink circles). **c** The axons of these ipRGCs are glutamatergic and VP-positive, and light stimulation can affect their secretion of VP. VP$^+$ ipRGCs showed synaptic co-localization with GRP (yellow) and VIP neurons, but VP$^+$ ipRGCs were not directly connected to AVP neurons[135]. **d** Some ipRGCs expressed GAD2 and could transmit GABA (solid purple circles) to the SCN regions. These GABAergic signals can excite or inhibit some SCN neurons, including VIP neurons, and maintain the homeostasis of the central rhythms[144]. AVP arginine vasopressin, GAD2 glutamic acid decarboxylase 2, GABA γ-aminobutyric acid, GRP gastrin releasing peptide, ipRGC intrinsically photosensitive retinal ganglion cell, RHT retinal hypothalamic tract, SCN supra-chiasmatic nucleus, VIP vasoactive intestinal peptide, VP vasopressin, PACAP pituitary adenylate cyclase-activating peptide.

increases the phase-shifting effects of light induction before a light pulse is provided in the early night rather than in the late night, suggesting that the inhibition of phase shift by extracellular GABA occurs mainly in the early night[142,143]. These data proved that changes in GABA in the SCN region are synchronized with non-visual light signals in the RGCs. Through GABAergic signaling, OPN4-expressing ipRGCs can also preserve circadian stability. Some ipRGCs co-expressing Gad2 and OPN4 can transmit GABAergic signals to the SCN to inhibit excessive light entrainment, and neurons receiving these GABAergic signals contain some VIP neuronal subsets[144]. Correspondingly, VIP neurons maintain the regular operation of the circadian rhythm, and inhibiting VIP neurons leads to increased phase shift[145].

Therefore, the optical signal mediated by OPN4 in ipRGCs is transmitted to the SCN via RHT. Light entrainment is mainly determined by glutamatergic transmitters and supplemented by multiple neuropeptides in the SCN to adjust the phase shift and intensity (Fig. 3). Simultaneously, GABAergic neurotransmitters may act as inhibitors in this terminal region. These inputs may prevent unnecessary adjustments of the master circadian clock in the SCN by external environmental light. Notably, the expression time of neuropeptides does not match the timing of the phase shift caused by it (such as PACAP)[126]. Considering that neuropeptides need to undergo an extended length of RHT

(mice: ~10 mm; rat: >20 mm) after they are synthesized from the cell body to the SCN region, they are transported only about ~140 mm per day along axons[84,146]. Therefore, when researching circadian rhythms, it would be interesting to look into the rate of transmission in the RHT and the rhythm of these neuropeptides' expression in the retina.

**Contribution of OPN4 to circadian rhythms in birds.** The central biological clock system of the bird is formed by the hypothalamus, retina, and pineal gland[147]. The non-visual photosensitization in GUCY1* chickens, a null mutation chicken model that causes blindness at hatching, is more complex than that in rd/rd cl mice. When blocking the input of light signals from the head, blocking the perception of light by opsins in the SCN and pineal gland directly reduces the effect of light drive on circadian rhythms[6,148]. On this basis, GUCY1* chickens showed a feeding rhythm disruption after enucleation[6]. This finding implies that the retina plays an essential role in maintaining the circadian rhythm in chickens. Additionally, when only hypothalamic photoreceptors were retained, GUCY1* chickens were still able to maintain a brief circadian phase shift to adapt to the next light-dark cycle, indicating that photoreceptors from the hypothalamus may play a role in light regulation of circadian feeding

behavior[148]. Although the evidence presented above does not exclude the possibility that other opsins have non-visual effects, the unique modulation of eating rhythm in GUCY1* chicken is particular to the light with wavelengths near the maximum absorption peak of OPN4. Another noteworthy example under light stimulation is that the pupillary light responses of chickens follow a circadian rhythm comparable to that of mammals[149]. It has been observed that GUCY1* chickens can maintain the circadian rhythm of pupillary light responses and reach maximum sensitivity at 480 nm[150]. Considering the light-absorbing properties of OPN4 and the distinct nonvisual photosensitivity function, the above findings strongly indicate that chicken retinal OPN4 regulates circadian rhythm. Notably, the photosensitivity function of OPN4 in the chicken retina may be more complex than is currently known. Chicken horizontal cells express OPN4x, which controls the release of GABA and regulates the membrane potential of photoreceptors following photosensitive activation[43]. Although this function is oriented more towards vision modulation, it cannot be excluded that OPN4x-expressing HCs may also affect the non-visual photosensitivity of OPN4-expressing retinal ganglion cells (RGCs) through signaling crosstalk.

The pineal gland of birds shows a robust melatonin secretion rhythm in *vivo* and in *vitro*. Monochromatic blue light (480 nm) can advance the phase of the rhythm-negative regulatory genes and inhibit the mRNA levels of *Cry1* and *Aanat* (a key enzyme in melatonin synthesis) in the pineal gland, both in vivo and in vitro[151,152]. Since the specific membrane receptors for melatonin are distributed in the SCN region, melatonin can act directly on the SCN in an endocrine form to regulate the clock rhythms in the SCN[153]. Compared with the pineal gland, the chicken retina is a relatively independent organ in the circadian rhythm, and pinealectomy does not alter the circadian oscillations in the retina[154]. The main effect of monochromatic blue light on the retinal circadian clock is to delay the phase of OPN4 rather than the phase shifts of clock genes or the mRNA levels of *Aanat*[147]. The SCN is the primary retinorecipient hypothalamic structure in birds[155]. When OPN4 in the chicken retina is excited by light, its non-visual light signals are mainly transmitted to the SCN. Then, the axons emitted from the SCN regulate downstream nuclei, such as the PVN and the infundibular nucleus (similar to the mammalian arcuate nucleus)[156]. It has been demonstrated that the hypothalamic appetite-related genes show a circadian rhythm[157]. Is it possible that ipRGCs expressing OPN4 might indirectly regulate the appetite of broilers through their neural projections to the SCN? Further investigation of the relationship between the non-visual photosensitive function of OPN4 and the feeding rhythm will help answer this question.

### Contribution of OPN4 to circadian rhythms in teleost fish.

Although the light-sensing mechanism of an extraretinal photoreceptor is unclear, it may represent the most basic approach to light-sensing[44,158]. The eye and pineal gland are the main central clock structures for zebrafish, which conduct autonomous oscillations, photoreception, and melatonin production[159]. The *eomesa*-expressing RGCs and pineal gland in zebrafish both express *Opn4.1* and *Opn4xb*[160,161]. When knocked out the *Opn4.1* and *Opn4xb* in the zebrafish, genes involved in phototransduction and tryptophan metabolism were significantly altered, resulting in increasing melatonin synthesis[64]. Meanwhile, by affecting the synaptic plasticity in hypothalamic neurons or directly acting on melatonin receptors distributed in the hypothalamic SCN, the light signals can control the circadian rhythms through OPN4 in zebrafish[162–165]. Therefore, OPN4 expressed in the retina and central nervous system constructs photosensitive sensing in zebrafish and regulates circadian rhythms through melatonin.

## Conclusions

OPN4 is a member of the G protein-coupled receptor family. Mammalian OPN4-expressing ipRGCs also express a variety of neurotransmitters and neuropeptides, which together with OPN4 regulate circadian rhythms. In contrast to mammals, teleost fish and birds have a more complicated system for controlling their circadian rhythms, and OPN4, which is expressed in the retina, brain, and pineal gland, is crucial for photosensitivity.

**Reporting summary**. Further information on research design is available in the Nature Portfolio Reporting Summary linked to this article.

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

## Acknowledgements
This work was supported by the National Natural Science Foundation of China (31972632, 31873000, and 31572474) and the Natural Science Foundation of Beijing Municipality (6192012).

## Author contributions
Conceptualization, J.C., D.P.; writing—original draft, D.P.; supervision, Z.W., Y.C.; project administration, J.C., Y.C. All authors have read and agreed to the published version of the manuscript.

## Competing interests
The authors declare no competing interests.
