## [Peer Review file · Communications Biology]

Reviewers' comments:

Reviewer #1 (Remarks to the Author):

The authors of this article have collated a vast amount of research to discuss the importance of melanopsin across multiple vertebrate species. Such a comparison will be a useful tool for researchers. The authors focus on mammalian melanopsin which is by far the most widely studied, but also make comparisons to that in fish and birds. They focus mostly on melanopsin expression, melanopsin phototransduction and also its role in photoentrainment.

General comments

With the current title I would find it a very useful resource if the authors were able to comment on additional vertebrates classes (ie reptiles and amphibians). You briefly mention discovery in xenopus and highlight it in table 1, but there is no additional information on our current understanding. Two additional short sections highlighting our current knowledge on these classes (both in terms of melanopsin expression, and maybe highlighting the complexity of light induced entrainment in these animals) would make the review article feel more complete under the current title.

I think it would also be important to highlight to the readers that ipRGCs also relay information from rod and cone photoreceptors to the SCN and that these photoreceptors can influence photoentrainment too.

A few preferences on terminology used throughout.

'non-visual perception', assuming you mean the same thing, typically it seems to be referred to as non-image forming visual pathways'. 'Blue light' as 'shorter wavelength light' as colour has connotations to cone photoreception, indeed 'white light' can activate melanopsin as much as 'blue light'. "Photoentrainment" is often used to explain entrainment due to light as opposed to "non-visual light entrainment (line 79).

Specific comments/considerations.

First Paragraph – lines 41-43. Perhaps it would be useful for the general audience to give a couple of examples of 'non-visual perception'.

Line 43 – Current references are from around when M1 cells were found. New techniques in mice now show M1-M6 and push the percentage in mice up a bit more. ~1-5% in mice – Sand review PMID 22480975, likewise there are more recent proportions for humans – see the mure review PMID: 33841306

Line 50 – It is unclear if you are talking about the long and short isoforms, or the x and m lineages. Please be specific for the naive reader.

Line 55- small grammatical change should 'OPN4 is a minor difference' be 'OPN4 shows some minor differences'

Line 57 – Would 'collate' be better than 'synthesize'.

I appreciate that the authors have commented on Gq and melanopsin (line 62) and suggested reviews for those keen to read. I am not sure I fully understand the following sentence (line63-65) as rods and cones both possess vitamin A based photopigments but show a completely different transduction pathway. If you are commenting on melanopsin phototransduction it might be worth to make a comment on the possible bi/tri-stability of melanopsin (see review <https://doi.org/10.3389/fopht.2023.1174255>)

Line 83, I was unable to find evidence for mammals losing OPN4x in favour for their nocturnal niche from the references mentioned. Are you able to find evidence for this.

Line 97, would be a good place to refer to fig 1.

Fig1: If the data is available it would be good to know if the expression of OPN4x and OPN4m overlapped exhibited distinct brain regions. Likewise if the same can be said for splice variants in fish. It might also be interesting to see insert of the retina for the three animals showing sparse melanopsin expression in the RGC layer (in ipRGCs) but also in a subset of horizontal cells in fish.

Line 120: Phrasing. As far as I am aware melanopsin (OPN4) are g-protein coupled receptors and so all will show classic 7 transmembrane alpha helices. So perhaps rephrase into Melanopsin is a g-protein coupled receptor which binds to.... '11-cis retinal'.

Line 122 is a little confusing, and may require a little bit more explanation to explain the complicated potential tristability of melanopsin. Possibly referring to rod/cone phototransduction where light causes a conformational change of 11-cis retinal to all trans retinal which activates the g-protein cascade.

Line 136: It seems a little bit of a leap from explaining the phototransduction cascade within ipRGCs, to the downstream targets of ipRGCs, in the OPN and SCN respectively. Indeed depending when light activates ipRGCs (and not necessarily via melanopsin) the circadian system will react differently.

Lines 152-4 To avoid confusion by a naive reader it might be beneficial to put function next to brain region, ie ipRGCs directly innervate the SCN (the centre for orchestrating circadian rhythms) and the OPN (important for PLR).

Line 163: Circadian rhythms can be found independently in multiple tissues. The SCN orchestrates these rhythms.

Line 191: not fully clear what you are referring to here, maybe some reference. Or is this what you go on to discuss below.

210-216: I may have misinterpreted this section, however are you just trying to say the strength of glutamatergic signalling to the SCN may vary between species. Or are you trying to show there is a difference in signalling between nocturnal and diurnal species. If the later I'm not sure there is sufficient evidence here to show that.

219: Not sure what you mean by nerve rhythm phase.

230-232: This might want some rephrasing as it sounds like in this study they manipulated c-fos and observed phase shifting behaviour. However they were merely recording c-fos levels under different lighting paradigms.

The section on potential effects of OPN4 photosensitive function on human health, may require some caveats. Blue light will stimulate multiple photoreceptors and may not be directly melanopsin mediated. Also it is important to distinguish what is mediated by light and what is due to circadian disruption. A lot of the negative effects reported here seem to be more related to circadian disruption than to light at night per say. Altogether, I am not sure whether this final section fits well with the rest of the review article.

Reviewer #2 (Remarks to the Author):

The review entitled "Melanopsin-mediated optical entrainment regulates circadian rhythm in vertebrates" by Deng Pan et al. investigates the role of the nonvisual blue opsin, melanopsin (OPN4) in the mechanisms of photic adjustment of circadian rhythms in vertebrates from teleost fish and birds to mammals with special focus on the neurobiological base and the molecular components.

This article attempts to make an original contribution regarding the key role played by melanopsin in retinal and extraocular photoreception and regulation of circadian rhythms in addition to other non-image forming tasks by light. Nevertheless, some concerns have been arisen by this reviewer to improve this manuscript as follows:

Please check the writing all through the text since many paragraphs are too long and lack inclusion of the corresponding verb.

Title, please consider write "circadian rhythm" in plural

Abstract, 1st paragraph. It is suggested:

"Melanopsin (OPN4) is a light-sensitive protein that plays a vital role in the regulation of circadian rhythms and other nonvisual functions."

Introduction, line 59, please include the verb in the following phrase: "We then highlight the mechanisms by which the non-visual photosensitization of OPN4"...after OPN4

Please, re write the question asked in the sentence going from line 310 to 312. It is not clear the meaning of it.

In addition to the very important role played by OPN4 in the photoreception responsible for setting the biological clock, it has been shown that this blue opsin also regulates a number of other nonvisual

tasks such as those related to a number of non-image forming functions as mentioned in the text. Indeed, a body of evidence in both chickens and rodents, has been shown to strongly imply OPN4 in the control of pupillary light reflexes even in the absence of visual photoreceptors. Please see Lucas et al 2003 and Valdez et al 2009.

In Contribution of OPN4 to circadian rhythms in birds: There is an article by Valdez et al 2015 IOVS showing the circadian control of the pupillary light responses in blind birds with highest sensitivity to light of 480 nm, that it would be useful to be considered.

This review also shed light on the photocascade mechanisms operated by OPN4, the ipRGCs and the neurotransmitter/neuropeptides involved and their retinoprojection to the SCN, highlighting the dorso/ventral organization.

With regards the photocascade, a couple of articles in mammals and birds, shows the involvement of GABA release in OPN4-expressing retinal neurons, please see Morera et al 2016 showing Opn4x-horizontal cells responding to blue light and Sonoda et al. 2020 describing a subset of Opn4-expressing ipRGCs in mice projecting to nonimage forming areas of the brain.

RESPONSE TO REVIEWERS' COMMENTS
(COMMSBIO-23-2346A-R1)

Dear Editor and Reviewers,

I'm very glad to hear from you, and thank you for your work in dealing with my manuscript. All the comments are valuable and very helpful for improving the quality of our manuscript, as well as the important guiding to our study. We have carefully checked and revised the manuscript according to the suggestions [red font color represented the revised text in the revised manuscript]. Now, I'm submitting here with a revised manuscript (R1).

Sincerely yours,

Jing Cao, Ph.D.

Laboratory of Anatomy of Domestic Animal, National Key Laboratory of Veterinary

Public Health and Safety, College of Veterinary Medicine,

China Agricultural University,

Haidian, Beijing, 100193, China

E-mail: caojing315@126.com

Reviewer #1 (Remarks to the Author):

1. General comments

1.1 With the current title I would find it a very useful resource if the authors were able to comment on additional vertebrates classes (ie reptiles and amphibians). You briefly mention discovery in xenopus and highlight it in table 1, but there is no additional information on our current understanding. Two additional short sections highlighting our current knowledge on these classes (both in terms of melanopsin expression, and maybe highlighting the complexity of light induced entrainment in these animals) would make the review article feel more complete under the current title.

Response: Thank you for your suggestion. We have added additional details about the OPN4 gene expression in amphibians and reptiles. Please see Lines 105-120 and Fig. 1 in the revised manuscript (R1). However, there is insufficient data on whether or how OPN4 regulates circadian rhythm in amphibians and reptiles.

To prevent repetition, we have removed information regarding the distribution of OPN4 in the brains of vertebrates from Table 1, because this part of the content is already well-represented in the updated Figure 1. The revised Table 1 has provided a concise summary of the current knowledge on the localization and function of OPN4 in peripheral organs.

1.2 I think it would also be important to highlight to the readers that ipRGCs also relay information from rod and cone photoreceptors to the SCN and that these photoreceptors can influence photoentrainment too.

Response: Thank you for your suggestion. We have added this section to the revised manuscript and briefly discuss the contribution of cone and rod photoreceptors to circadian photoentrainment. Please see Lines 206-216 in the revised manuscript (R1).

1.3 A few preferences on terminology used throughout.

‘non-visual perception’, assuming you mean the same thing, typically it seems to

**be referred to as non-image forming visual pathways'. 'Blue light' as 'shorter wavelength light' as colour has connotations to cone photoreception, indeed 'white light' can activate melanopsin as much as 'blue light'.
"Photoentrainment" is often used to explain entrainment due to light as opposed to "non-visual light entrainment (line 79).**

Response: Thank you for your correction, we have made the following changes:

1) Incorrect term expression "nonvisual perception" has been modified to "non-image forming visual pathways" . Please see Lines 43-44 and Line 48 in the revised manuscript (R1).

2) We have replaced "Blue light" in some sentences as you suggested. Please see Line 132, 159, 331, 333, and 353 in the revised manuscript (R1). In other places, we have kept some similar descriptions for the corresponding wavelengths of light. Please see Line 218, 344, and 350 in the revised manuscript (R1).

On the activation of OPN4, most of the current experimental conditions are not performed in dim light. The illuminance and light power ($> 1 \mu\text{W}$) required for OPN4 activation appear to be easily achievable in a standard white light environment^{1,2}. However, in some reference papers (especially regarding birds), the group exposed to blue light demonstrated a distinct regulation of the clock gene compared to the group exposed to white light in low illuminance (15 lux)³. Therefore, we have retained "Blue light" in this part of the results to prevent differences in OPN4 activation between blue and white light.

3) We have replaced "non-visual light entrainment" with "photoentrainment". Please see Line 83 in the revised manuscript (R1).

References

1. Foster, R. G., Hughes, S. & Peirson, S. N. Circadian photoentrainment in mice and humans. *Biology* **9**, 180 (2020).
2. Ratnayake, K., Payton, J. L., Lakmal, O. H. & Karunaratne, A. Blue light excited retinal intercepts cellular signaling. *Sci. Rep.* **8**, 10207 (2018).
3. Bian, J., Wang, Z., Dong, Y., Cao, J. & Chen, Y. Effect of monochromatic light on the circadian

clock of cultured chick retinal tissue. *Exp. Eye Res.* **194**, 108008 (2020).

2. Specific comments/considerations.

2.1 First Paragraph – lines 41-43. Perhaps it would be useful for the general audience to give a couple of examples of ‘non-visual perception’.

Response: Thank you for your suggestion. We have included examples in the sentences and given the relevant references. Please see Lines 45-46 in the revised manuscript (R1).

2.2 Line 43 – Current references are from around when M1 cells were found. New techniques in mice now show M1-M6 and push the percentage in mice up a bit more. ~1-5% in mice – Sand review PMID 22480975, likewise there are more recent proportions for humans – see the mure review PMID: 33841306.

Response: Thank you for your correction. We have corrected the description of the number of ipRGCs and added references. Please see Line 47 in the revised manuscript (R1).

2.3 Line 50 – It is unclear if you are talking about the long and short isoforms, or the x and m lineages. Please be specific for the naive reader.

Response: Thank you for your suggestion. We have replaced the sentence with "It was also found that there are two OPN4m splice variants in mice and humans, the short (OPN4-S) and long (OPN4-L) isoforms ...". Please see Lines 53-54 in the revised manuscript (R1).

2.4 Line 55- small grammatical change should ‘OPN4 is a minor difference’ be ‘OPN4 shows some minor differences’

Response: Thank you for your correction. We have revised this grammatical error. Please see Line 59 in the revised manuscript (R1).

2.5 Line 57 – Would ‘collate’ be better than ‘synthesize’.

Response: Thank you for your correction. The word has been changed in the revised manuscript. Please see Line 61 in the revised manuscript (R1).

2.6 I appreciate that the authors have commented on Gq and melanopsin (line 62) and suggested reviews for those keen to read. I am not sure I fully understand the following sentence (line63-65) as rods and cones both possess vitamin A based photopigments but show a completely different transduction pathway. If you are commenting on melanopsin phototransduction it might be worth to make a comment on the possible bi/tri-stability of melanopsin (see review <https://doi.org/10.3389/fopht.2023.1174255>)

Response: Thank you for your comment. Although OPN4 phototransduction relies on a vitamin A-based chromophore, our presentation could easily lead readers to believe that only the chromophore of ipRGCs is vitamin A-based. To correct this misrepresentation, we have rewritten the sentence: "In this article, the G_{q/11} pathway in the OPN4-mediated phototransduction was mainly described due to its widespread presence in vertebrate ipRGCs". Please see Lines 67-69 in the revised manuscript (R1). In addition, we have added evidence for G_{q/11}-dependent phototransduction pathways in humans, mice, birds, reptiles, and fish OPN4 as references to reflect their "widespread presence"¹⁻⁴.

Thank you very much for the link to the review article. We briefly described the conformation of OPN4 in phototransduction (see the section on **Light activation of OPN4 in the retina**) but did not highlight the significance of the tristability for the unique phototransduction properties of melanopsin. Please see Lines 140-145 in the revised manuscript (R1).

References

1. Xue, T. et al. Melanopsin signalling in mammalian iris and retina. *Nature* **479**, 67-73 (2011).
2. Bailes, H. J. & Lucas, R. J. Human melanopsin forms a pigment maximally sensitive to blue light (lambda_{max} approximately 479 nm) supporting activation of G_(q/11) and G_(i/o) signalling cascades. *Proc. Biol. Sci.* **280**, 20122987 (2013).

3. Crowe-Riddell, J. M. et al. Phototactic tails: Evolution and molecular basis of a novel sensory trait in sea snakes. *Mol. Ecol.* **28**, 2013-2028 (2019).
4. Ramos, B. C., Moraes, M. N., Poletini, M. O., Lima, L. H. & Castrucci, A. M. From blue light to clock genes in zebrafish ZEM-2S cells. *PLoS One* **9**, e106252 (2014).

2.7 Line 83, I was unable to find evidence for mammals losing OPN4x in favour for their nocturnal niche from the references mentioned. Are you able to find evidence for this.

Response: Thank you for your comment. The evidence has been supplemented that mammals lose some of their opsins during a "nocturnal bottleneck" in evolution. Kaas *et al.* describe the development of visual function in mammals while escaping the "nocturnal bottleneck" over 66 million years ago¹. Borges *et al.* demonstrated the loss of mammalian visible proteins during this process². Of note, the above evidence only suggests that the loss of OPN4x coincides with the adaptation of mammals to the nocturnal ecological niche, and further studies are needed to determine the role of OPN4x loss in this process.

Additionally, we replace this sentence to describe the relevant evidence.

"Mammals lost OPN4x during evolution and chromosomal re-arrangements, which accompanied mammal adaptation to the nocturnal niche". Please see Lines 87-88 in the revised manuscript (R1).

References

1. Kaas, J. H., Qi, H.-X. & Stepniewska, I. Escaping the nocturnal bottleneck, and the evolution of the dorsal and ventral streams of visual processing in primates. *Philosophical Transactions of the Royal Society B: Biological Sciences* **377**, 20210293 (2021).
2. Borges, R. et al. Adaptive genomic evolution of opsins reveals that early mammals flourished in nocturnal environments. *BMC Genomics* **19**, 121 (2018).

2.8 Line 97, would be a good place to refer to fig 1.

Response: Thank you for your suggestion. The text has been added according to your

suggestion. Please see Line 102 in the revised manuscript (R1).

2.9 Fig1: If the data is available it would be good to know if the expression of OPN4x and OPN4m overlapped exhibited distinct brain regions. Likewise if the same can be said for splice variants in fish. It might also be interesting to see insert of the retina for the three animals showing sparse melanopsin expression in the RGC layer (in ipRGCs) but also in a subset of horizontal cells in fish.

Response: Thank you for your suggestion. We have described the localization of OPN4m and OPN4x in the retina, brain, and pineal gland in vertebrates (Fig. 1). It should be noted that there is less evidence from in situ hybridization, immunofluorescence, etc., in amphibians and reptiles. We have only summarized widely accepted evidence to ensure our report is accurate. Please see Figure 1 in the revised manuscript (R1)

2.10 Line 120: Phrasing. As far as I am aware melanopsin (OPN4) are g-protein coupled receptors and so all will show classic 7 transmembrane alpha helices. So perhaps rephrase into Melanopsin is a g-protein coupled receptor which binds to.... '11-cis retinal'.

Response: Thank you for your suggestion. We have revised this sentence with "OPN4 is a G protein-coupled receptor with 11-*cis* retinal as a covalently bound protonated Schiff base (PSB11)". Please see Lines 140-141 in the revised manuscript (R1).

2.11 Line 122 is a little confusing, and may require a little bit more explanation to explain the complicated potential tristability of melanopsin. Possibly referring to rod/cone phototransduction where light causes a conformational change of 11-cis retinal to all trans retinal which activates the g-protein cascade.

Response: Thank you for your suggestion. About your above comments on the tristability of OPN4 (Please refer to 2.6), we have added the works of Emanuel *et al.* to reflect the uniqueness of OPN4 in the phototransduction process^{1,2}. Please see Lines 140-145 in the revised manuscript (R1).

References

1. Emanuel, A. J. & Do, M. T. Melanopsin tristability for sustained and broadband phototransduction. *Neuron* **85**, 1043-1055 (2015).
2. Emanuel, A. J. & Do, M. T. H. The multistable melanopsins of mammals. *Frontiers in Ophthalmology* **3**; 10.3389/fopht.2023.1174255 (2023).

2.12 Line 136: It seems a little bit of a leap from explaining the phototransduction cascade within ipRGCs, to the downstream targets of ipRGCs, in the OPN and SCN respectively. Indeed depending when light activates ipRGCs (and not necessarily via melanopsin) the circadian system will react differently.

Response: Thank you for your comment. To avoid confusion for the reader, we have changed this ambiguous sentence to "Its phosphorylation process preferentially interacts with G protein-coupled receptor, kinase 2/3 (GRK2/3), preventing OPN4-expressing ipRGCs from generating sustained action potentials after light stimulation". Please see Lines 157-159 in the revised manuscript (R1).

2.13 Lines 152-4 To avoid confusion by a naive reader it might be beneficial to put function next to brain region, ie ipRGCs directly innervate the SCN (the centre for orchestrating circadian rhythms) and the OPN (important for PLR).

Response: Thank you for your suggestion. In order to avoid any misinterpretation of the function of these nuclei, we have re-written the section as "OPN4-induced non-visual photosensitive signals can target numerous nuclei, including the SCN, intergeniculate leaflet, ventral lateral geniculate nucleus, and olivary pretectal nucleus (OPN). The SCN is the center for orchestrating mammalian circadian rhythms, while the OPN is essential for regulating the pupillary light reflex". Please see Lines 174-177 in the revised manuscript (R1).

2.14 Line 163: Circadian rhythms can be found independently in multiple tissues. The SCN orchestrates these rhythms.

Response: Thank you for your correction. It was an explaining error, and we have corrected it according to your suggestion. Please see Lines 185-186 in the revised manuscript (R1).

2.15 Line 191: not fully clear what you are referring to here, maybe some reference. Or is this what you go on to discuss below.

Response: Thank you for your correction. This sentence has been rewritten "In addition to this approach, OPN4-positive ipRGCs can rely on self-synthesized neurotransmitters and neuropeptides to more directly and rapidly affect the SCN master clock". Please see Lines 223-225 in the revised manuscript (R1).

2.16 210-216: I may have misinterpreted this section, however are you just trying to say the strength of glutamatergic signalling to the SCN may vary between species. Or are you trying to show there is a difference in signalling between nocturnal and diurnal species. If the later I'm not sure there is sufficient evidence here to show that.

Response: Thank you for your comment. We aim to convey a dissimilarity in the potency of glutamatergic signalling for controlling circadian rhythms in the two species. Although Nile rats and Syrian hamsters vary in the quantity and arrangement of ipRGCs isoforms, they possess the same OPN4-positive ipRGCs projection network, circadian behavior, and expression patterns of most clock genes¹. Therefore, OPN4-mediated circadian regulation may be similar between nocturnal and diurnal species. However, significant dissimilarities exist in the level of response to light in the two species. When NMDA was used to mimic the photoentrainment², there was a substantial difference in the amount of NMDA needed for the maximum phase shift to transpire in Nile rats and Syrian hamsters³. We hypothesize that there may be variations in the strength of glutamatergic signalling tolerance among species with distinct activity patterns. These differences aid in safeguarding diurnally active species against excessive light stimulation⁴.

References

1. Challet, E. Minireview: Entrainment of the suprachiasmatic clockwork in diurnal and nocturnal mammals. *Endocrinology* **148**, 5648-5655 (2007).
2. Mintz, E. M., Marvel, C. L., Gillespie, C. F., Price, K. M. & Albers, H. E. Activation of NMDA receptors in the suprachiasmatic nucleus produces light-like phase shifts of the circadian clock in vivo. *J. Neurosci.* **19**, 5124-5130 (1999).
3. Novak, C. M. & Albers, H. E. N-Methyl-D-aspartate microinjected into the suprachiasmatic nucleus mimics the phase-shifting effects of light in the diurnal Nile grass rat (*Arvicanthis niloticus*). *Brain Res.* **951**, 255-263 (2002).
4. Fogo, G. M., Shuboni-Mulligan, D. D. & Gall, A. J. Melanopsin-containing ipRGCs are resistant to excitotoxic injury and maintain functional non-image forming behaviors after insult in a diurnal rodent model. *Neuroscience* **412**, 105-115 (2019).

2.17 219: Not sure what you mean by nerve rhythm phase.

Response: Thank you for your correction. we have replaced "nerve rhythm phase" with "the peak of the SCN activity rhythm" to avoid ambiguity. Please see Line 254 in the revised manuscript (R1).

2.18 230-232: This might want some rephrasing as is sounds like in this study they manipulated cfos and observed phase shifting behaviour. However they were merely recording c-fos levels under different lighting paradigms.

Response: Thank you for your comment. We have rewritten this sentence: "It has been proven that if light triggers c-Fos expression mainly in the dorsomedial area of the SCN, mice do not experience any rhythmic phase shifts". Please see Lines 264-266 in the revised manuscript (R1).

2.19 The section on potential effects of OPN4 photosensitive function on human health, may require some caveats. Blue light will stimulate multiple photoreceptors and may not be directly melanopsin mediated. Also it is important to distinguish what is mediated by light and what is due to circadian

disruption. A lot of the negative effects reported here seem to be more related to circadian disruption than to light at night per say. Altogether, I am not sure whether this final section fits well with the rest of the review article.

Response: Thank you for your comment. As you commented, we may be exaggerating the non-visual photosensitizing effects of OPN4. The section has been deleted in the revised manuscript (R1).

Reviewer #2 (Remarks to the Author):

1. Please check the writing all through the text since many paragraphs are too long and lack inclusion of the corresponding verb.

Response: Thank you for your correction. We rechecked for grammatical problems and made corrections in the revised manuscript.

2. Title, please consider write “circadian rhythm” in plural

Response: Thank you for your suggestion. The title has been modified according to your suggestion. Please see Line 2 and 4 in the revised manuscript (R1).

3. Abstract, 1st paragraph. It is suggested:

“Melanopsin (OPN4) is a light-sensitive protein that plays a vital role in the regulation of circadian rhythms and other nonvisual functions.”

Response: Thank you for your suggestion. The **Abstract** has been modified according to your suggestion. Please see Lines 27-28 in the revised manuscript (R1).

4. Introduction, line 59, please include the verb in the following phrase: “We then highlight the mechanisms by which the non-visual photosensitization of OPN4”...after OPN4

Response: Thank you for your correction. We have added the verb "mediates" in this sentence. Please see Line 63 in the revised manuscript (R1).

5. Please, rewrite the question asked in the sentence going from lane 310 to 312. It is not clear the meaning of it.

Response: Thank you for your correction. We have rewritten the question: "Is it possible that ipRGCs expressing OPN4 might indirectly regulate the appetite of broilers through their neural projections to the SCN?". Please see Lines 357-339 in the revised manuscript (R1).

6. In addition to the very important role played by OPN4 in the photoreception responsible for setting the biological clock, it has been shown that this blue opsin also regulates a number of other nonvisual tasks such as those related to a number of non-image forming functions as mentioned in the text. Indeed, a body of evidence in both chickens and rodents, has been shown to strongly imply OPN4 in the control of pupillary light reflexes even in the absence of visual photoreceptors. Please see Lucas et al 2003 and Valdez et al 2009.

Response: Thank you for your suggestion. Since the manuscript focuses on the OPN4-mediated regulation of circadian rhythms, we have supplemented the evidence for pupillary constriction that you suggested in the Introduction.

Additionally, a reviewer suggested we should include examples when introducing non-visual perception. Therefore, we have referred to these significant works when introducing examples of non-image-forming functions of OPN4. Please see Line 45 in the revised manuscript (R1).

7. In Contribution of OPN4 to circadian rhythms in birds: There is an article by Valdez et al 2015 IOVS showing the circadian control of the pupillary light responses in blind birds with highest sensitivity to light of 480 nm, that it would be useful to be considered.

Response: Thank you for your suggestion. We have added this critical evidence to the revised manuscript, which broadens our discussion on the contribution of OPN4 to circadian rhythms in birds. Please see Lines 331-336 in the revised manuscript (R1).

8. This review also shed light on the photocascade mechanisms operated by OPN4, the ipRGCs and the neurotransmitter/neuropeptides involved and their retinoprojection to the SCN, highlighting the dorso/ventral organization.

With regards the photocascade, a couple of articles in mammals and birds, shows the involvement of GABA release in OPN4-expressing retinal neurons, please see Morera et al 2016 showing Opn4x-horizontal cells responding to blue light and Sonoda et al. 2020 describing a subset of Opn4-expressing ipRGCs in mice

projecting to nonimage forming areas of the brain.

Response: Thank you for your suggestion. Based on the original manuscript, we have added the papers on OPN4x to the part "**Contribution of OPN4 to circadian rhythms in birds**". We believe that this study demonstrates the photosensitivity of OPN4x in chicken HCs, indicating that birds have a more complex OPN4 photoregulatory system than mammals. Please see Lines 336-342 in the revised manuscript (R1).

In the original manuscript, we have discussed the work of Sonoda *et al.* 2020, please see Lines 300-302 in the revised manuscript (R1).

REVIEWERS' COMMENTS:

Reviewer #1 (Remarks to the Author):

We thank the Author for taking on our suggestions. A huge kudos for adding our current knowledge on both reptiles and amphibians.

A much more complete and wholesome read. A couple minor additional things but otherwise I am very happy with your responses to the comments.

Comment number 2. Thanks for addressing the influence of rod and cone photoreceptors on the SCN. You have cover that amply. However I would recommend for perhaps the lay reader just a line saying that ipRGCs relay rod and cone information to the brain. To a naïve reader they may otherwise interpret that ipRGCs, rods and cones as three distinct pathways. (See Schmidt - Functional and morphological differences among intrinsically photosensitive retinal ganglion cells - or - Intrinsic and extrinsic light responses in melanopsin-expressing ganglion cells during mouse development)

Line 37, In the abstract, can be removed as you have now removed the section on humans.

Line 129-130 I assume you mean OPN4m or OPN4x in the relevant places

Lines 264-266, I am a little cautious about using the word proven. Personally I interpret the results of this study slightly differently. The SCN shows a daily variation in c-fos expression throughout the 24 hours day (even in constant darkness). The phrasing also might be interpreted that c-fos activity in the dorsomedial SCN inhibits phase shifting.

Line 281: Would probably change "promote mice to enter a next experimental light/dark cycle." to "promotes (near instantaneous) re-entrainment to the new light/dark cycle"

RESPONSE TO REVIEWERS' COMMENTS
(COMMSBIO-23-2346B R2)

Dear Editor and Reviewers,

I'm very glad to hear from you, and thank you for your work in dealing with my manuscript. All the comments are valuable and very helpful for improving the quality of our manuscript, as well as the important guiding to our study. We have carefully checked and revised the manuscript according to the **Reviewers' Comments** and **Final Revision Instructions** [red font color represented the revised text in the revised manuscript]. Now, I'm submitting here with a revised manuscript (R2).

Sincerely yours,

Jing Cao, Ph.D.

Laboratory of Anatomy of Domestic Animal, National Key Laboratory of Veterinary
Public Health and Safety, College of Veterinary Medicine,
China Agricultural University,
Haidian, Beijing, 100193, China
E-mail: caojing315@126.com

Reviewer #1 (Remarks to the Author):

1. Comment number 2. Thanks for addressing the influence of rod and cone photoreceptors on the SCN. You have cover that amply. However, I would recommend for perhaps the lay reader just a line saying that ipRGCs relay rod and cone information to the brain. To a naïve reader they may otherwise interpret that ipRGCs, rods and cones as three distinct pathways. (See Schmidt - Functional and morphological differences among intrinsically photosensitive retinal ganglion cells - or - Intrinsic and extrinsic light responses in melanopsin-expressing ganglion cells during mouse development)

Response: Thank you for your comment. We have complemented the work of Schmidt *et al.* in our manuscript and separately demonstrated that ipRGCs can act as relays to transmit rod and cone information to the brain. Please see Lines 207-210 in the revised manuscript (R2).

2. Line 37, In the abstract, can be removed as you have now removed the section on humans.

Response: Thank you for your correction. We have revised the expression of this sentence. Please see Lines 37-38 in the revised manuscript (R2).

3. Line 129-130 I assume you mean OPN4m or OPN4x in the relevant places

Response: Thank you for your comment. OPN4a mediates light-seeking behavior in zebrafish larvae, and this OPN4 splice variant can be classified as OPN4m. OPN4x2 differs from OPN4a in losing its original circadian expression rhythm in cells with clock gene mutations. Therefore, we modify this sentence to "Evidence for functional partitioning suggests that OPN4m mediates the light-seeking behavior in larvae distributed in the preoptic area, whereas OPN4x regulates circadian rhythms in the SCN". Please see Lines 129-131 in the revised manuscript (R2).

4. Lines 264-266, I am a little cautious about using the word proven. Personally, I interpret the results of this study slightly differently. The SCN shows a daily

variation in c-fos expression throughout the 24 hours day (even in constant darkness). The phrasing also might be interpreted that c-fos activity in the dorsomedial SCN inhibits phase shifting.

Response: Thank you for your correction. We have modified this sentence to avoid exaggerated results and potential misunderstandings. Please see Lines 267-269 in the revised manuscript (R2).

Although c-Fos is just used as a marker of neuronal activity in this evidence and is not directly related to the phase-shifting mechanism under light entrainment, we retain the word "c-Fos" to accurately reflect the evidence we cited.

5. Line 281: Would probably change "promote mice to enter a next experimental light/dark cycle." to "promotes (near instantaneous) re-entrainment to the new light/dark cycle"

Response: Thank you for your suggestion. We have revised this sentence as you recommend. Please see Line 285 in the revised manuscript (R2).